# Quality Characteristics and Antioxidant Activity of Fresh Noodles Formulated with Flour-Bran Blends Varied by Particle Size and Blend Ratio of Purple-Colored Wheat Bran

**Gyuna Park [1], Hyejin Cho [1], Kyeonghoon Kim [2] and Meera Kweon [1,3,*]**

[1] Department of Food Science and Nutrition, Pusan National University, Busan 46241, Korea; honeychuchu@pusan.ac.kr (G.P.); tyu117@pusan.ac.kr (H.C.)

[2] Wheat Team, National Institute of Crop Science, Rural Development Administration, Jeonju 55365, Korea; k2h0331@korea.kr

[3] Kimchi Institute, Pusan National University, Busan 46241, Korea

[*] Correspondence: meera.kweon@pusan.ac.kr; Tel.: +82-51-510-2716

**Abstract:** This study explored the noodle-making performance of flour blends with different particle sizes and blending ratios of purple-colored wheat bran and their antioxidant properties. The bran particle size was reduced using an ultra-centrifugal mill equipped with 1, 0.5, and 0.2 mm sieves. The damaged starch and swelling capacity of the bran were analyzed. Quality of the flour-bran blends at different blending ratios was analyzed by solvent retention capacity (SRC). Noodles made from the blends and their corresponding antioxidant activities were examined. The damaged starch and swelling capacity of bran were higher for smaller particles than for larger particles. Water and sodium carbonate SRC values of blends increased as the bran particle size decreased. The smaller the bran particles incorporated in the cooked noodles, the greater firmness and springiness measured. The antioxidant activity of noodles made with blends reflected better embedding of the small particles of bran than the large particles into noodle sheets. Small bran particles significantly enhanced noodles' quality and antioxidant activity at higher blending ratios than large bran particles. Particle size reduction of bran enhanced the noodle-making performance of flour blended with purple-colored wheat bran; this could increase the utilization of bran to produce noodles with health benefits.

**Keywords:** purple-colored wheat; wheat bran; particle size; blending ratio; fresh noodles; antioxidant properties

## 1. Introduction

Wheat (*Triticum aestivum*) is a typical crop used in various food products worldwide. Wheat bran, a by-product generated during wheat milling, usually accounts for 14–19% of the grain weight and consists of 37–52% total dietary fiber [1,2]. Bran has nutritional properties and provides many functional benefits, including antioxidant activity [3,4]. The main antioxidative components in wheat bran are polyphenols (mainly phenolic acids) [5]. In particular, purple- or black-colored wheat contains more anthocyanin and polyphenol compounds in the wheat bran and aleurone layers than does common wheat, providing a more significant nutritional advantage [6,7].

Despite these benefits, bran in whole wheat flour (WWF) presents challenges to producing high-quality bakery or noodle products. Wheat bran negatively influences the qualities of bread dough by physically interfering with the gluten matrix of the dough [8,9]. It also destroys the starch–gluten matrix of the dough and leads to lower firmness of cooked noodles and higher loss of solids in cooking water with increased bran content [10]. Aravind et al. [11] reported that wheat bran in pasta negatively affects the cooking and sensory quality of pasta due to increased cooking loss and swelling index, leading to unfavorable sensory acceptance and appearance. Additionally, whole wheat products

are of less attractive quality and have lower sensory acceptance than do refined wheat products [12,13]. Accordingly, the current intake of whole wheat grain products is still much lower than the recommended level worldwide.

Noodles are a wheat-based food widely used as a meal substitute [14]. The development of noodles made from WWF can be an effective way to increase the demand for low-calorie healthy foods and promote high-fiber foods. Noodles made with WWF provide enhanced nutritional benefits compared to those with regular flour, but they are not appealing to many consumers because of their negative sensory properties such as texture and taste. When bran is blended with refined wheat flour for noodle production, it is challenging to increase the blending ratio of bran because of its large particle size and high water absorption compared to wheat flour itself; this hinders dough development. Numerous methods such as fermentation [15], enzyme treatment [16,17], germination [18], particle size control by milling [19,20], and physical treatments including hydration, autoclaving, and freezing [5,21,22] have been investigated to solve these issues.

Among these methods, particle size reduction of bran by milling can alter its physical and functional properties [23–25], causing structural rather than chemical modification [20]. Particle size reduction of wheat bran can improve the quality of wheat-based products. Fine bran is less destructive to dough-mixing and allows better gluten network formation than does coarse bran [26]. Particle size reduction of wheat bran has a minimal effect on dough characteristics, leading to high sensory acceptability of the resulting noodles [10,27]. However, the effect of the bran particle size on the volume of breads is somewhat contradictory due to additional effects of bran composition and type [9,19].

The majority of the research on bran particle size was performed for brans with different particle sizes prepared with different pulverizers [24] or by sifting with various sieves after pulverization [3,28]. Memon et al. [28] reported the direct influence of WWF particle size on the distribution of phenolic acid, carbohydrate, protein, crude fiber, ash, crude fat, and moisture in the three commercial wheat varieties when the WWF was fractionated by shifting with a series of five sieves. In comparison, it is expected that bran milled with the same pulverizer equipped with different sieves separately might differ in only particle size, not in the distribution of nutritional components. However, the research on the effect of bran particle size segregated from the effect of the distribution of nutritional components in bran is rare and worth investigating. The National Institute of Crop Science in Korea recently released a purple-colored wheat cultivar, "Ariheuk", as value-added wheat for enhancing health benefits. Avarzed et al. [29] reported that the purple-colored wheat bran has significantly higher total phenolic and anthocyanin content and antioxidant activity than common wheat bran. In terms of this new cultivar, it is worth investigating the fresh noodle-making performance of flour-bran blends by increasing the blending ratio of bran and reducing its particle size using the same pulverizer equipped with different sieves.

The present study explored the effect of particle size reduction of purple-colored wheat bran on the hydration properties, solvent retention capacity (SRC), and dough-mixing properties of flour-bran blends with different bran blending ratios. In addition, the fresh noodle-making performance of the blends and the antioxidant activity of the noodles were assessed.

## 2. Materials and Methods

### 2.1. Materials

Korean domestic wheat flour milled from the cultivar "Keumkang" and purple-colored wheat bran obtained from the cultivar "Ariheuk" used in this study were supplied by the National Institute of Crop Science in Korea. To produce noodles, salt (Samyang, Seoul, Korea) was purchased from a local market. To analyze SRC, sodium carbonate (Duksan, Seoul, Korea) was used as extra-pure-grade reagent.

### 2.2. Size Reduction of Purple-Colored Wheat Bran and Measurement of Bran Particle Size

To measure the particle size of the bran before milling, the bran (300 g) was separated with three sieves (DAIHAN Scientific, Wonju, Korea) with 2.0, 1.0, and 0.5 mm openings, respectively. The bran passed through each sieve was weighed, and the percentage of each fraction was: 5% ≤ 0.5 mm, 0.5 mm < 24% < 1.0 mm, 1.0 mm < 68% < 2.0 mm, 2.0 mm ≤ 3%. To reduce the bran particle size, bran was milled using an ultra-centrifugal mill (POWTEQ FM200, Beijing, China), which was equipped with a 12-tooth rotor and three-ring individually inserted sieves (1.0, 0.5, and 0.2 mm), at a centrifugal speed of 15,000 rpm. Brans of different sizes were obtained and labeled as L (1.0 mm), M (0.5 mm), and S (0.2 mm). The original bran sample was labeled VL because it was much larger than L, M, and S bran samples.

The particle size of the milled bran was measured using a laser-scattering particle size analyzer (Beckman Coulter LS 13 320, Fullerton, CA, USA) equipped with a vacuum delivery system for dry samples and the maximum particle size limit of 2000 μm.

### 2.3. Analysis of Physicochemical Properties of Purple-Colored Wheat Bran

The moisture content of the bran samples was measured according to Method 44-15.02 [30]. The total starch content of only VL bran sample and the damaged starch content of all bran samples were measured using the Megazyme Total Starch Assay Kit (K-TSTA-100A, Megazyme International, Wicklow, Ireland) and the Starch Damage Assay Kit (K-SDAM) by Method 76-31.01 [30], respectively.

The swelling capacity of the bran samples was determined according to the method reported by Jacobs et al. [20] with slight modifications. Bran (750 mg) was soaked in deionized water (7.5 mL) in a 10 mL graduated cylinder for 60 min to absorb water and swell. The volume of the swollen bran is referred to as the swelling capacity.

### 2.4. Preparation of Flour-Bran Blends with Different Blending Ratios

"Keumkang" flour and "Ariheuk" bran were blended at ratios of 9:1, 7:3, and 5:5 and denoted by F90-B10, F70-B30, and F50-B50, respectively. In addition, four flour-bran blends with different blending ratios were prepared with four bran sizes (S, M, L, and VL). Flour alone was designated as F100-B0.

### 2.5. Solvent Retention Capacity of Flour-Bran Blends

SRC analysis of the flour-bran blends was performed according to Method 56-1.02 [30]. SRC in only water and sodium carbonate solutions was tested because the SRC values of wheat bran in lactic acid and sucrose solutions tend to be under or overestimated [31]. Flour-bran blend (5 g) was added to pre-weighed 50 mL conical tubes. Two solvents, 25 g of distilled water, and 5% ($w/w$) sodium carbonate solution were prepared separately in each tube. Each solution was poured into a tube containing the flour-bran blend, and each tube was shaken every 5 min for 20 min to sufficiently disperse the flour-bran blend. The suspension of each flour-bran blend was centrifuged at $1000 \times g$ for 15 min using a centrifuge (LaboGene1248, Gyrozen Inc., Daejeon, Korea), and the supernatant was discarded. The tube containing the pellet was weighed, and the SRC value (%) was calculated according to Method 56-1.02.

### 2.6. Preparation of Fresh White-Salted Noodles Formulated with the Flour-Bran Blends

Fresh noodles were prepared using the method described by Moon et al. [32]. The flour-bran blend (100 g) was placed in the bowl of a micro mixer (National Manufacturing Inc., Lincoln, NE, USA), and 2 g of salt and 30–43.3 g of distilled water, based on the water absorption capacity measured by water SRC value of each blend, was added (Table 1). The flour-bran blend was mixed with salt and water for 15 min, and the prepared dough was placed in a plastic bag and rested for 30 min in a resting chamber (Phantom M301 Combi, Phantom Korea, Hanam, Korea) at 35 °C and 85% relative humidity. The rested dough was sheeted into thicknesses of 3.0, 2.0, and 1.5 mm continuously with a noodle-maker

(SN-88, Samwoo Industrial Co., Daegu, Korea) and then cut (4.0 mm width) to produce fresh noodles. One strand of each noodle from the fresh noodles prepared with flour-bran blends at different blending ratios and bran sizes was aligned in parallel together and photos were taken to observe the appearance of fresh noodles.

**Table 1.** Ingredients and formula used for preparing fresh noodles made with flour-bran blends at different ratios.

| Ingredients | Formula (g) | | | |
|---|---|---|---|---|
| | F100-B0 [1] | F90-B10 | F70-B30 | F50-B50 |
| Flour | 100.0 | 90.0 | 70.0 | 50.0 |
| Wheat bran | 0.0 | 10.0 | 30.0 | 50.0 |
| Water | 30.0 | 32.0 | 37.4 | 43.3 |
| Salt | 2.0 | 2.0 | 2.0 | 2.0 |

[1] F100-B0, Keumkang flour; F90-B10, F70-B30, and F50-B50: blends of Keumkang flour and Ariheuk bran at ratios of 9:1, 7:3, and 5:5, respectively.

### 2.7. Color Measurements of Fresh Noodle Sheets and Cooked Noodles

Before cutting a noodle sheet of 1.5 mm thickness into noodles, the color of the noodle sheet was measured using a colorimeter (CR-20, Konica Minolta, Tokyo, Japan). The color parameters of brightness (L*), redness (a*), and yellowness (b*) were repeatedly measured five times and calculated as average values.

To measure the color of cooked noodles, 15 g of fresh noodles were placed in 500 mL of boiling water and boiled for 15 min according to the methods reported by Moon et al. [32] and Wang and Kweon [33]. The cooking water was drained separately into a beaker to measure turbidity. The color of the cooked noodles was measured for five noodle strands arranged side by side. The remaining cooked noodles were used to measure texture.

### 2.8. Measurement of Turbidity of Cooking Water

The turbidity of the cooking water after boiling noodles was measured by absorbance using a spectrophotometer (X-ma 6100PC, Human Corporation, Seoul, Korea) at 675 nm.

### 2.9. Analysis of Textural Property of Fresh and Cooked Noodles

As a representative method to measure the texture of fresh noodles, the extensibility of the noodles was measured using a texture analyzer (CT3, Brookfield, Middleboro, MA, USA), and the measurement conditions were as follows: test mode, tension; pretest speed, 2 mm/s; test speed, 3.3 mm/s; probe, Kieffer rig (TA-KF); target value, 20 mm. The average value was calculated from ten measurements.

The texture of the cooked noodles was also analyzed using a texture analyzer on five strands of noodles. The measurement conditions were as follows: test mode, TPA; pretest speed, 2 mm/s; test speed, 1 mm/s; probe, Asian noodle rig (TA 7); deformation%, 70. Firmness, cohesiveness, springiness, and chewiness were measured, and the average value was calculated from five measurements.

### 2.10. Measurement of Total Polyphenol Content of Fresh Noodles

The total polyphenol content (TPC) was determined by a slight modification of the method described by Yu and Beta [34]. Fresh noodles were freeze-dried and ground using a grinder (WSG-9100, Joong San Co., Seoul, Korea). Ground noodles (2 g) were extracted twice with 20 mL of 80% methanol in a 50 mL tube by shaking for 90 min with a rotator (CN/VM-80, Miulab, Hangzhou, China). The noodle mixture was sonicated in an ice-filled sonicator (LK-U105, LK Lab Korea, Namyangju, Korea) at 40 kHz for 30 min in the dark. After centrifuging the noodle mixture at $12,000\times g$ and 4 °C for 15 min, the supernatant was filtered with 90 mm filter paper (Qualitative Filter Paper No.2, ADVATEC, Tokyo, Japan) and stored at −20 °C. The TPC of the noodles was determined using Folin–Ciocalteu reagent (Sigma, St. Louis, MO, USA). The noodle extract (0.2 mL) was oxidized with

10-times-diluted Folin–Ciocalteu reagent (1.5 mL) for 5 min and neutralized with 1.5 mL sodium carbonate solution (60 g/L). After 90 min, the absorbance of the reacted noodle extract was measured at 725 nm against a blank of distilled water using a spectrophotometer (X-ma 6100PC, Human Corporation, Seoul, Korea). Gallic acid (Sigma, St. Louis, MO, USA) was used as the standard. The TPC of fresh noodles was expressed as mg GAE (gallic acid equivalent)/100 g.

### 2.11. Measurement of Total Anthocyanin Content of Fresh Noodles

The total anthocyanin content (TAC) of fresh noodles was determined using the method described by Yu and Beta [34]. Acidified methanol (methanol: 1.0 N HCl = 85:15 (*v*/*v*), Ph = 1) was used to extract anthocyanins from the noodles. The procedure to react the noodle mixture with Folin–Ciocalteu reagent was the same as that used for TPC. Absorbance was measured at 535 nm. Cyanidin-3-glucoside was used as a standard in an 80% methanol blank. The results were expressed as mg C3GE (cyaniding-3-glucoside equivalent)/100 g sample.

### 2.12. Antioxidant Activity of Fresh Noodles—ABTS Radical Scavenging Capacity

Antioxidant activity as indicated by the ABTS radical scavenging capacity of fresh noodles was determined by a slight modification of the method of Yu and Beta [34]. First, 10 mL of ABTS reagent was diluted with approximately 990 mL of distilled water. Then, antioxidant activity of phenolic and anthocyanin extracts (50 μL) from the noodles was evaluated by adding 1.85 mL of the diluted ABTS reagent. Absorbance was measured at t = 30 min. The absorbance of the reaction solution was measured at 734 nm using a spectrophotometer (X-ma 6100PC, Human Corporation, Seoul, Korea) after 30 min on an 80% methanol blank. For absorbance at t = 0 min, 1.85 mL of the diluted ABTS reagent was added to 100 μL of 80% methanol. A standard curve was generated based on different Trolox concentrations vs. %ABTS decolorization, and the ABTS value was expressed as μL TE (Trolox equivalents)/100 g sample.

### 2.13. Antioxidant Activity of Fresh Noodles—DPPH Radical-Scavenging Capacity

Antioxidant activity as indicated by DPPH radical-scavenging capacity of fresh noodles was determined by a slight modification of the method of Yu and Beta [34]. A 55 μL/L DPPH reagent was prepared by dilution in 100% methanol. After adding 3.9 mL of DPPH solution to 0.1 mL of noodle extract, the absorbance was measured at t = 30 min at 515 nm using a spectrophotometer (X-ma 6100PC, Human Corporation, Seoul, Korea) on an 80% methanol blank. After adding 3.9 mL of DPPH solution to 0.1 mL of 80% methanol, the absorbance was measured at t = 0 min. A standard curve was generated based on different Trolox concentrations vs. % DPPH scavenging activity. DPPH value was expressed as μL TE/100 g sample.

### 2.14. Statistical Analysis

All data were obtained from measurements in at least triplicate. Differences between samples (significance threshold, $p < 0.05$) were analyzed by ANOVA and Tukey's HSD test using SPSS 22.0 (SPSS Inc., Armonk, New York, NY, USA).

## 3. Results and Discussion

### 3.1. Particle Size of Purple-Colored Wheat Bran Milled with an Ultra-Centrifugal Mill

The particle size distributions of the reduced sizes of the brans are shown in Figure 1. All bran samples (L, M, and S) with reduced particle size showed a unimodal shape with small humps. These are notably distinct in particle size, demonstrating an appropriate selection for exploring the effect of particle size reduction of bran in this study. The bran samples exhibited a significantly increased portion of smaller particles due to particle size reduction. The calculated mean particle sizes of the L, M, and S bran samples were 584.0,

255.2, and 123.1 μm, respectively. The particle sizes of the brans were significantly different, potentially resulting in different noodle-making performance [10,27,35].

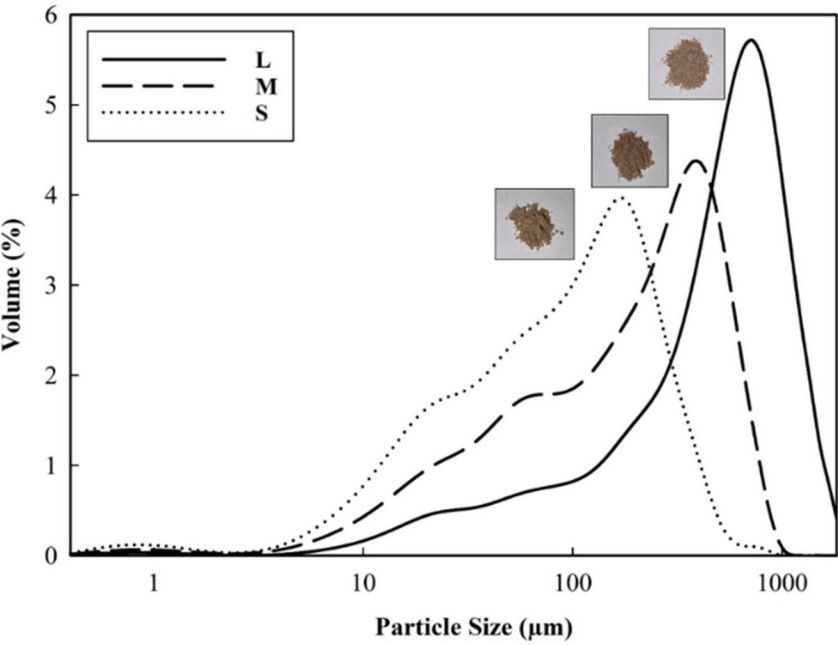

**Figure 1.** The particle size distribution of wheat bran with size reduction determined by laser diffraction. L, M, and S indicate the brans with large, medium, and small particle sizes.

### 3.2. Physicochemical Properties of Purple-Colored Wheat Bran

Physicochemical properties of bran samples are presented in Table 2. The moisture content of the samples was 9.0–12.2% and decreased as the bran size decreased. This decrease was due to the moisture loss by frictional heat produced in milling wheat bran because more frictional heat is produced when making bran of smaller sizes [36].

**Table 2.** Moisture and damaged starch contents and swelling capacities of purple-colored wheat bran with different sizes.

| Bran Size | Moisture Content (%) | Damaged Starch Content (%) | Swelling Capacity (%) |
|---|---|---|---|
| VL [1] | 12.2 ± 0.0 [d(2)] | 1.9 ± 0.0 [a] | 58.7 ± 0.0 [a] |
| L | 11.9 ± 0.2 [c] | 2.7 ± 0.0 [b] | 74.2 ± 2.0 [b] |
| M | 10.2 ± 0.1 [b] | 3.0 ± 0.1 [c] | 80.4 ± 0.8 [bc] |
| S | 9.0 ± 0.1 [a] | 4.0 ± 0.1 [d] | 85.3 ± 3.5 [d] |

[1] VL, L, M, and S indicate the brans with very large, large, medium and small particle sizes. [2] Results are expressed as mean ± SD. Values with the different letters within the same column are significantly different ($p < 0.05$) according to Tukey's HSD test.

The VL bran contained approximately 23.9% of total starch. The damaged starch content of the bran samples is shown in Table 2; it increased with a decrease in bran size. The damaged starch content of VL bran was 1.9%, and that of S bran was 4.0%, which could affect water absorption, dough property, and noodle quality of the bran samples. Fu [14] reported that the increased damaged starch in wheat flour increased water absorption and decreased dough-mixing stability. The noodles formulated with the flour-bran blend with S bran required increased water for dough development. With increasing damaged starches in flour, noodles may increase cooking loss and become gummier and less firm [37].

The swelling capacity of the bran significantly increased with decreasing bran size (Table 2). The swelling capacity of VL bran was 58.7%, and that of S bran was 85.3%,

reflecting an increase in hydration properties as the particle size decreased. Even L bran showed a dramatic increase in swelling capacity, confirming the significant effect of bran size reduction on swelling power. Sanz Penella et al. [8] reported that decreased wheat bran size could increase the farinograph water absorption value through more water interaction via hydrogen bonding in the fiber structure. However, other studies have shown the opposite impact: larger particles of wheat bran have greater swelling capacity than smaller particles in terms of hydration properties [20,38,39]. In addition, the hydration properties of bran vary by differences in the composition of bran depending on the wheat cultivar and growing environment [19,40] and in the distribution of chemical components between the bran layers [41,42].

*3.3. Solvent Retention Capacity Values of Flour-Bran Blends*

The SRC values of the flour-bran blends are listed in Table 3. Flour alone without bran exhibited 60.9% water SRC and 78.7% sodium carbonate SRC values. For the flour-bran blends with VL bran, the water SRC values were 65.1 for F90-B10, 76.0 for F70-B30, and 88.6% for F50-B50. With an increase in the bran blending ratio, the water SRC values of the blends increased significantly. For the same blending ratio of bran, the water SRC values of the blends increased (e.g., from 76.6 to 78.8% for F70-B30) as the bran size decreased from L to S. Cai et al. [19], De Bondt et al. [24], and Habuš et al. [39] showed an inconsistent result on water-holding/retention capacity of bran due to differences in milling methods used to reduce the bran particle size and in wheat classes tested. De Bondt et al. [24] explained an increase in water SRC upon particle size reduction by an increase in specific surface area.

**Table 3.** Moisture content and solvent retention capacity values of flour-bran blends with different bran sizes and blend ratios.

| Flour-Bran Blend | Bran Size | SRC (%) | |
| :---: | :---: | :---: | :---: |
| | | **Water** | **Sodium Carbonate** |
| F100-B0 [1] | - | 60.9 ± 0.2 [a(2)] | 78.7 ± 0.1 [a] |
| F90-B10 | VL [3] | 65.0 ± 0.3 [b] | 82.7 ± 0.0 [b] |
| | L | 65.4 ± 0.1 [bc] | 83.2 ± 0.4 [bc] |
| | M | 65.8 ± 0.1 [bc] | 84.1 ± 0.3 [bc] |
| | S | 66.5 ± 0.1 [c] | 84.7 ± 0.0 [c] |
| F70-B30 | VL | 76.0 ± 0.4 [d] | 94.6 ± 0.4 [d] |
| | L | 76.6 ± 0.1 [d] | 95.8 ± 0.1 [d] |
| | M | 76.9 ± 0.2 [d] | 97.5 ± 0.4 [e] |
| | S | 78.8 ± 0.3 [e] | 99.3 ± 0.1 [f] |
| F50-B50 | VL | 88.6 ± 0.1 [f] | 106.0 ± 0.2 [g] |
| | L | 89.5 ± 0.1 [fg] | 110.8 ± 0.3 [h] |
| | M | 89.8 ± 0.0 [fg] | 113.2 ± 0.4 [i] |
| | S | 90.3 ± 0.4 [g] | 113.8 ± 0.3 [i] |

[1] F100-B0, Keumkang flour; F90-B10, F70-B30, and F50-B50: blend of Keumkang flour and Ariheuk bran at ratios of 9:1, 7:3, and 5:5, respectively. [2] Results are expressed as mean ± SD. Values with the different letters within the same column are significantly different ($p < 0.05$) according to Tukey's HSD test. [3] VL, L, M, and S indicate the brans with very large, large, medium, and small particle sizes.

The sodium carbonate SRC values were 82.7 for F90-B10, 94.6 for F70-B30, and 106.0% for F50-B50. As the blending ratio increased, the sodium carbonate SRC values of the blends also increased, showing a similar trend to the water SRC values. Within the same blending ratio, the sodium carbonate SRC values of the blends increased as the bran size decreased. As a representative example, the sodium carbonate SRC values of F70-B30 increased from 95.8% for L bran to 99.3% for S bran. Water SRC and sodium carbonate SRC are indicators of water absorption and the contribution of damaged starch, respectively. The sodium carbonate SRC solution (5% $w/w$) has a pH $\approx$ 12.0, which is above the pK of starch hydroxyl groups. Under this condition, damaged starch can be easily solvated by sodium

carbonate solution and shows exaggerated swelling [31]. Sodium carbonate SRC values were affected significantly by reducing the bran size due to the increased contribution of damaged starch [43,44]. Overall, the SRC results suggested an increased amount of water required for making noodle dough with the flour-bran blends upon increasing the bran blending ratio.

### 3.4. Appearance and Texture of Fresh Noodles

The appearance of fresh noodles prepared from flour-bran blends with different blending ratios and bran sizes is shown in Figure 2. The noodles became dark as the bran blending ratio increased and had a smooth surface as the bran size reduced. In addition, the noodles with the smallest size of bran appeared the bran to be embedded uniformly by adhering firmly in the noodles.

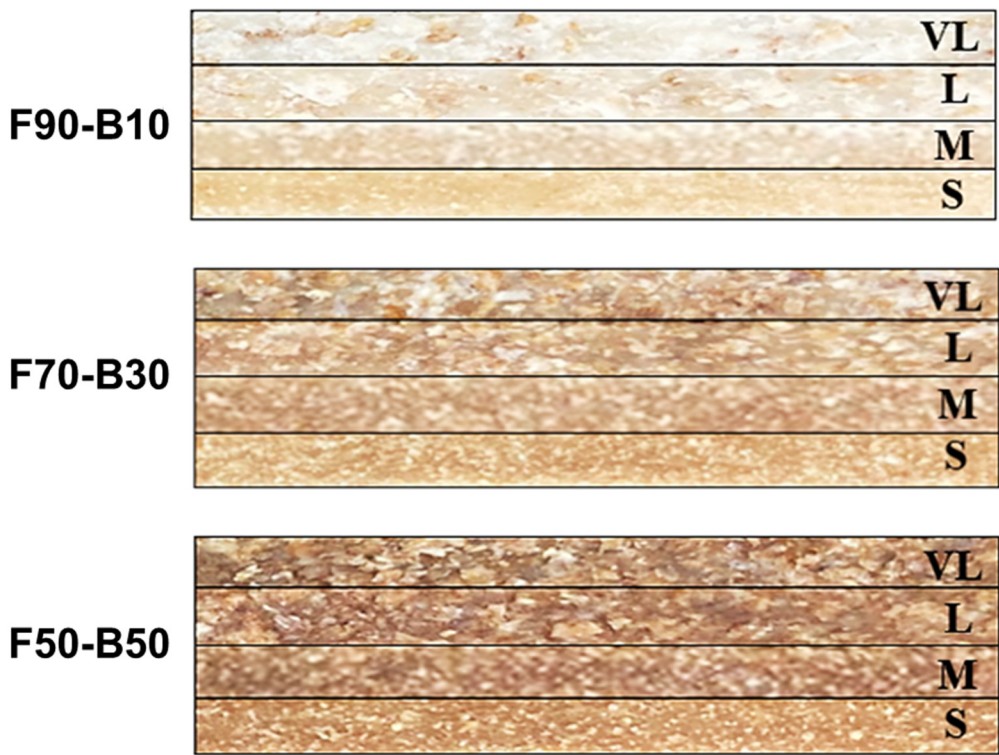

**Figure 2.** Appearance of fresh noodles prepared with flour-bran blends at different blending ratios and bran sizes. VL, L, M, and S indicate the brans with very large, large, medium, and small particle sizes.

The resistance to extension (R), extensibility (E), and R/E values of fresh noodles measured for textural properties are shown in Table 4. Gliadins contribute to dough viscosity and extensibility, and glutenin imparts elasticity. The R/E ratio of fresh noodles represents a balance between dough strength and extensibility, indicating gluten network strength [45]. An appropriate balance between dough viscosity and elasticity is essential for well-developed doughs [46]. The fresh noodles prepared with flour alone (F100-B0) exhibited 1.37 N resistance, 14.76 mm extensibility, and 0.093 R/E ratios. The resistance values of fresh noodles prepared with flour-bran blends with VL bran were 1.18 for F90-B10, 0.83 for F70-B30, and 0.51 N for F50-B50, decreasing significantly with increasing blending ratio. Within the same blending ratio, the resistance of the noodles increased (e.g., from 1.00 to 1.18 N for F70-B30) as bran size decreased from L to S ($p < 0.05$). The extensibility values of the blends with VL bran were 11.20 for F90-B10, 10.84 for F70-B30, and 10.49 mm for F50-B50, decreasing slightly with an increase in the bran blending ratio. Within the same blending ratio, the extensibility of noodles with blends decreased slightly or negligibly

when bran size was reduced from L to S (e.g., from 9.67 to 9.58 mm for F70-B30). The R/E values of fresh noodles with blends containing VL bran were 0.106 for F90-B10, 0.077 for F70-B30, and 0.048 for F50-B50, decreasing significantly with an increase in the blending ratio ($p < 0.05$). Within the same blending ratio, the R/E ratios of noodles with blends increased significantly with decreasing bran size from L to S (e.g., from 0.103 to 0.123 for F70-B30) ($p < 0.05$).

**Table 4.** Resistance (R), extensibility (E), and R/E values of flour-bran blends with different bran sizes and blend ratios.

| Flour-Bran Blend | Bran Size | Resistance (N) | Extensibility (mm) | R/E |
|---|---|---|---|---|
| F100-B0 [1] | - | 1.37 ± 0.04 [e(2)] | 14.76 ± 0.07 [f] | 0.093 ± 0.003 [cd] |
| F90-B10 | VL [3] | 1.18 ± 0.05 [de] | 11.20 ± 0.03 [e] | 0.106 ± 0.006 [de] |
| | L | 1.20 ± 0.03 [de] | 10.08 ± 0.15 [bc] | 0.119 ± 0.002 [ef] |
| | M | 1.32 ± 0.03 [e] | 9.81 ± 0.07 [ab] | 0.134 ± 0.003 [fg] |
| | S | 1.42 ± 0.03 [e] | 9.78 ± 0.06 [ab] | 0.145 ± 0.003 [g] |
| F70-B30 | VL | 0.83 ± 0.03 [bc] | 10.84 ± 0.09 [de] | 0.077 ± 0.003 [bc] |
| | L | 1.00 ± 0.05 [cd] | 9.67 ± 0.09 [ab] | 0.103 ± 0.006 [de] |
| | M | 1.16 ± 0.10 [de] | 9.64 ± 0.09 [a] | 0.120 ± 0.013 [ef] |
| | S | 1.18 ± 0.08 [de] | 9.58 ± 0.05 [a] | 0.123 ± 0.011 [efg] |
| F50-B50 | VL | 0.51 ± 0.01 [a] | 10.49 ± 0.01 [cd] | 0.048 ± 0.001 [a] |
| | L | 0.55 ± 0.02 [a] | 9.50 ± 0.07 [a] | 0.058 ± 0.002 [ab] |
| | M | 0.71 ± 0.04 [ab] | 9.53 ± 0.01 [a] | 0.074 ± 0.006 [bc] |
| | S | 0.74 ± 0.03 [abc] | 9.52 ± 0.03 [a] | 0.078 ± 0.004 [bc] |

[1] F100-B0, Keumkang flour; F90-B10, F70-B30, and F50-B50: blend of Keumkang flour and Ariheuk bran at ratios of 9:1, 7:3, and 5:5, respectively. [2] Results are expressed as mean ± SD. Values with the different letters within the same column are significantly different ($p < 0.05$) according to Tukey's HSD test. [3] VL, L, M, and S indicate the brans with very large, large, medium, and small particle sizes.

Our results for resistance and extensibility of noodle dough sheets within the same ratio of flour-bran blend showed trends similar to other publications. Bran particle size distribution of WWF influences gluten network formation [47]. The effect of the bran particle size on the dough rheological property is somewhat contradictory due to the effects of the milling method and bran composition. Smaller bran particles resulted in higher resistance of bread dough than larger particles [48], and reduced particle size of wheat bran by micronization decreased dough extensibility [49]. The dough containing fine bran exhibited higher dough strength after resting for 180 min in the farinograph and extensograph evaluations, owing to the quick recovery of the gluten network of the dough [50]. Sudha et al. [51] reported that the dough extensibility of WWF decreased with an increase in the bran level. Overall, it can be speculated that the textural properties of fresh noodle sheets might reflect stronger gluten strength in the noodles with flour-bran blends containing the decreased size of bran. Özboy and Köksel [52] explained that coarse brans occupy more space and result in a more detrimental effect on the gluten network formation in dough by decreasing connectivity.

### 3.5. Turbidity of Cooking Water and Textural Characteristics of Cooked Noodles

The color values of the fresh dough sheet and cooked noodles prepared from flour-bran blends with different blend ratios and bran sizes are shown in Table 5. As the bran blending ratio increased, the fresh dough sheet showed a decrease in brightness (L*) and an increase in redness (a*), indicating the significant influence of purple-colored bran. Bran size also affected the color of the dough sheet, which slightly decreased in brightness and increased in redness and yellowness (b*) with size reduction.

**Table 5.** Color of fresh noodle sheets and cooked noodles prepared from flour-bran blends with different bran sizes and blend ratios.

| Flour-Bran Blend | Bran Size | Uncooked | | | Cooked | | |
|---|---|---|---|---|---|---|---|
| | | L* | a* | b* | L* | a* | b* |
| F100-B0 [1] | - | $86.2 \pm 0.5$ [f(2)] | $2.7 \pm 0.1$ [a] | $15.3 \pm 0.8$ [d] | $74.4 \pm 0.3$ [h] | $(-0.8 \pm 0.1)$ [a] | $12.0 \pm 0.2$ [bcd] |
| F90-B10 | VL [3] | $73.5 \pm 1.8$ [e] | $3.8 \pm 0.3$ [ab] | $9.6 \pm 0.4$ [a] | $56.5 \pm 1.1$ [g] | $4.6 \pm 0.6$ [b] | $7.3 \pm 1.1$ [a] |
| | L | $72.0 \pm 1.2$ [de] | $4.8 \pm 0.4$ [bc] | $10.7 \pm 0.6$ [ab] | $54.8 \pm 0.7$ [fg] | $5.8 \pm 0.2$ [bc] | $8.9 \pm 0.3$ [ab] |
| | M | $67.2 \pm 1.9$ [de] | $6.0 \pm 0.5$ [cd] | $11.6 \pm 0.5$ [abc] | $52.2 \pm 0.4$ [f] | $6.5 \pm 0.3$ [bcd] | $10.4 \pm 0.4$ [abcd] |
| | S | $65.9 \pm 0.5$ [d] | $6.8 \pm 0.2$ [de] | $12.8 \pm 0.4$ [bcd] | $51.1 \pm 0.7$ [f] | $7.2 \pm 0.3$ [cde] | $10.9 \pm 0.2$ [abcd] |
| F70-B30 | VL | $57.8 \pm 1.3$ [c] | $6.4 \pm 0.4$ [cde] | $11.6 \pm 0.7$ [abc] | $45.4 \pm 1.4$ [e] | $6.7 \pm 0.6$ [cd] | $9.3 \pm 1.4$ [abc] |
| | L | $54.0 \pm 1.7$ [c] | $8.0 \pm 0.5$ [ef] | $13.2 \pm 0.6$ [bcd] | $44.2 \pm 0.4$ [de] | $7.4 \pm 0.4$ [cde] | $10.1 \pm 0.6$ [abc] |
| | M | $51.4 \pm 1.0$ [bc] | $8.5 \pm 0.1$ [fg] | $13.8 \pm 0.3$ [cd] | $41.3 \pm 0.4$ [cd] | $8.4 \pm 0.4$ [defg] | $12.3 \pm 1.1$ [bcde] |
| | S | $53.9 \pm 0.6$ [c] | $9.1 \pm 0.1$ [fg] | $14.7 \pm 0.3$ [cd] | $40.4 \pm 0.5$ [bcd] | $9.4 \pm 0.3$ [fg] | $14.0 \pm 0.6$ [de] |
| F50-B50 | VL | $46.0 \pm 2.1$ [a] | $8.9 \pm 0.2$ [g] | $13.6 \pm 0.2$ [cd] | $38.1 \pm 1.0$ [abc] | $7.7 \pm 0.2$ [cdef] | $9.7 \pm 0.5$ [ab] |
| | L | $41.7 \pm 0.6$ [a] | $9.7 \pm 0.1$ [g] | $13.9 \pm 0.2$ [cd] | $37.2 \pm 0.3$ [ab] | $8.8 \pm 0.3$ [efg] | $12.0 \pm 0.4$ [bcde] |
| | M | $41.4 \pm 0.6$ [a] | $9.5 \pm 0.2$ [fg] | $13.9 \pm 0.3$ [cd] | $35.3 \pm 0.3$ [a] | $9.4 \pm 0.2$ [fg] | $13.8 \pm 0.4$ [cde] |
| | S | $43.6 \pm 0.5$ [ab] | $9.9 \pm 0.1$ [g] | $14.6 \pm 0.2$ [cd] | $34.8 \pm 0.2$ [a] | $10.2 \pm 0.2$ [g] | $15.1 \pm 0.3$ [e] |

[1] F100-B0, Keumkang flour; F90-B10, F70-B30, and F50-B50: blend of Keumkang flour and Ariheuk bran at ratios of 9:1, 7:3, and 5:5, respectively. [2] Results are expressed as mean $\pm$ SD. Values with the different letters within the same column are significantly different ($p < 0.05$) according to Tukey's HSD test. [3] VL, L, M, and S indicate the brans with very large, large, medium, and small particle sizes.

The turbidity of the cooking water for the noodles prepared from the flour-bran blends is presented in Table 6. The fresh noodle prepared with flour alone (F100-B0) exhibited turbidity of 0.78 $\Delta$A hr$^{-1}$ g flour$^{-1}$ in the cooking water. The turbidity values of the cooking water for fresh noodles with the blends with VL bran were 1.09 for F90-B10, 1.30 for F70-B30, and 1.35 $\Delta$A hr$^{-1}$ g flour$^{-1}$ for F50-B50, increasing significantly up to the 30% blending ratio and leveling off with a further increase in the blending ratio. Within the same blending ratio, the turbidity in cooking water for the noodles with blends decreased significantly (e.g., from 0.69 to 0.62 $\Delta$A hr$^{-1}$ g flour$^{-1}$ for F70-B30) as bran size decreased from L to S ($p < 0.05$). The increased turbidity of the cooking water with an increasing blending ratio indicated that more solids leached from the noodles by loosening and weakening the gluten network structure. In contrast, the decreased turbidity of the cooking water with decreasing bran size reflected fewer solids leaching out from the noodles with a stronger gluten network structure. High cooking loss reflected the weaker gluten strength and lower degree of connectivity in the noodle structure [53].

The textural parameters of the cooked noodles made with the flour-bran blends are listed in Table 6. Cooked noodles prepared with flour alone exhibited 18.1 N firmness, 0.62 cohesiveness, 0.91 springiness, and 19.8 mJ chewiness. The firmness of cooked noodles prepared with the blends with VL bran decreased significantly from 13.9 N for F90-B10 to 4.4 N for F50-B50 with an increasing blending ratio ($p < 0.05$). Chen et al. [27] reported a decrease in hardness of cooked dry white Chinese noodles by increasing bran addition level, similar to our result. Within the same blending ratio of bran, the firmness of the noodles with blends increased significantly (e.g., from 8.8 to 22.0 N for F50-B50) as bran size decreased from L to S ($p < 0.05$). These increases were more significant in the noodles with a higher blending ratio than those with a lower blending ratio. Although the damaged starch content increased by decreasing bran size (Table 2), the noodles showed a decrease in turbidity of cooking water and an increase in firmness of cooked noodles, which might be a more significant contribution by gluten development than by damaged starch.

**Table 6.** Textural parameters of cooked noodles prepared from flour alone and flour-bran blends with different bran sizes and blend ratios.

| Flour-Bran Blend | Bran Size | Textural Parameter | | | | Turbidity of Cooked Water ($\Delta$A hr$^{-1}$ g flour$^{-1}$) |
|---|---|---|---|---|---|---|
| | | Firmness (N) | Cohesiveness | Springiness | Chewiness (mJ) | |
| F100-B0 [1] | - | 18.1 ± 0.9 [def(2)] | 0.62 ± 0.00 [c] | 0.91 ± 0.00 [c] | 19.8 ± 1.2 [e] | 0.78 ± 0.06 [e] |
| F90-B10 | VL [3] | 13.9 ± 0.2 [bcd] | 0.56 ± 0.02 [c] | 0.84 ± 0.01 [bc] | 15.5 ± 0.9 [de] | 1.09 ± 0.01 [f] |
| | L | 14.6 ± 0.7 [cde] | 0.57 ± 0.02 [c] | 0.86 ± 0.03 [bc] | 16.0 ± 1.5 [de] | 0.65 ± 0.01 [bcd] |
| | M | 16.2 ± 1.1 [de] | 0.57 ± 0.02 [c] | 0.90 ± 0.01 [c] | 16.6 ± 1.6 [de] | 0.46 ± 0.02 [a] |
| | S | 16.4 ± 0.9 [de] | 0.59 ± 0.02 [c] | 0.91 ± 0.02 [c] | 17.1 ± 1.0 [de] | 0.56 ± 0.00 [ab] |
| F70-B30 | VL | 9.5 ± 0.6 [abc] | 0.43 ± 0.03 [abc] | 0.74 ± 0.01 [abc] | 7.7 ± 0.5 [abc] | 1.30 ± 0.01 [g] |
| | L | 14.7 ± 0.7 [cde] | 0.49 ± 0.12 [bc] | 0.83 ± 0.05 [bc] | 13.1 ± 4.3 [cde] | 0.69 ± 0.02 [cde] |
| | M | 17.8 ± 0.5 [def] | 0.46 ± 0.02 [abc] | 0.85 ± 0.04 [bc] | 13.7 ± 0.8 [cde] | 0.59 ± 0.00 [bc] |
| | S | 19.3 ± 1.1 [ef] | 0.48 ± 0.02 [abc] | 0.86 ± 0.02 [bc] | 15.8 ± 0.6 [de] | 0.62 ± 0.00 [bc] |
| F50-B50 | VL | 4.4 ± 0.5 [a] | 0.24 ± 0.01 [a] | 0.57 ± 0.02 [a] | 2.0 ± 0.4 [a] | 1.35 ± 0.01 [g] |
| | L | 8.8 ± 0.7 [ab] | 0.29 ± 0.06 [ab] | 0.57 ± 0.09 [a] | 3.8 ± 0.9 [ab] | 0.74 ± 0.01 [de] |
| | M | 14.5 ± 1.3 [cde] | 0.31 ± 0.02 [ab] | 0.67 ± 0.03 [ab] | 7.0 ± 1.7 [abc] | 0.65 ± 0.01 [bcd] |
| | S | 22.0 ± 1.8 [f] | 0.31 ± 0.06 [ab] | 0.80 ± 0.02 [bc] | 10.5 ± 1.9 [bcd] | 0.63 ± 0.00 [bc] |

[1] F100-B0, Keumkang flour; F90-B10, F70-B30, and F50-B50: blend of Keumkang flour and Ariheuk bran at ratios of 9:1, 7:3, and 5:5, respectively. [2] Results are expressed as mean ± SD. Values with the different letters within the same column are significantly different ($p < 0.05$) according to Tukey's HSD test. [3] VL, L, M, and S indicate the brans with very large, large, medium, and small particle sizes.

The cohesiveness values of cooked noodles prepared with flour-bran blends with VL bran were 0.56 for F90-B10, 0.43 for F70-B30, and 0.24 for F50-B50. The springiness values of cooked noodles prepared from blends with VL bran were 0.84 for F90-B10, 0.74 for F70-B30, and 0.57 for F50-B50. The cohesiveness and springiness of cooked noodles with the blends significantly decreased with an increase in the blending ratio ($p < 0.05$). Among noodles with same blending ratio, the cohesiveness of noodles increased negligibly (e.g., from 0.29 to 0.31 for F50-B50) with increasing bran size from L to S. However, springiness of noodles with blends increased significantly (e.g., from 0.57 to 0.80 for F50-B50) as bran size decreased from L to S ($p < 0.05$). Niu et al. [26] reported that WWF noodles increased in springiness and cohesiveness with decreasing millfeed particle size.

The chewiness of cooked noodles prepared from the blends containing VL bran were 15.5 for F90-B10, 7.7 for F70-B30, and 2.0 mJ for F50-B50. Similar to the results for firmness, the chewiness of cooked noodles with the blends significantly decreased with increasing blending ratio ($p < 0.05$). Chen et al. [27] also reported a decrease in chewiness of cooked dry white Chinese noodles by increasing bran addition level. Within the same blending ratio, the chewiness of the noodles with blends increased significantly (e.g., 3.8 to 10.5 mJ for F50-B50) as bran size decreased from L to S ($p < 0.05$).

Overall, as the bran size decreased, the firmness, cohesiveness, springiness, and chewiness of cooked noodles increased while the turbidity of the cooking water decreased, demonstrating improved desirable attributes of noodle. This result was similar to that reported by Sozer et al. [54], who reported that the reduction of bran particle size in biscuits showed the highest elastic modulus value because overall strength was improved by better incorporating bran particles into the biscuit matrix. Hatcher et al. [35] also reported excellent cooking quality of noodles with a fine particle size of flour. Ma et al. [55] explained that a small particle size of flour increases the change in the gluten index and wet gluten content of the flour and increases the hardness, elasticity, adhesiveness, and chewiness of the noodles by increasing the network structure and water-holding capacity of the noodles. Small bran particles enhanced noodle quality more significantly at higher blending ratios. In the future, a sensory evaluation will be needed to correlate with the textural data measured by the instrument to confirm the eating quality.

### 3.6. Total Phenolic Compound and Total Anthocyanin Contents of Fresh Noodles Prepared with Flour-Bran Blends

The total phenolic compound and total anthocyanin contents of fresh noodles prepared from the flour-bran blends with different blend ratios and bran sizes are shown in Figure 3. Fresh noodles prepared with flour alone exhibited a total phenolic content of 69.9 mg GAE/100 g and total anthocyanin content of 0.31 mg C3GE/100 g. The total phenolic content (TPC) of fresh noodles with the blend with VL bran was 158.7 for F90-B10, 302.8 for F70-B30, and 482.8 mg GAE/100 g for F50-B50, increasing significantly and proportionally as the blending ratio increased ($p < 0.05$). Within the same blending ratio of bran, the total phenolic content of the noodles with blends increased significantly (e.g., from 328.8 to 380.1 mg GAE/100 g for F70-B30, and from 504.2 to 546.3 mg GAE/100 g for F50-B50) as bran size decreased from L to S. The TPC increase is possibly due to an increase in surface area to mass by particle size reduction and the increased extraction of phytochemicals [56]. Black-purple wheat varieties have higher total phenolic content than white varieties [57]. Avarzed et al. [29] also reported that purple-colored wheat bran contains higher total phenolic content than normal wheat bran, which is a similar trend to our result.

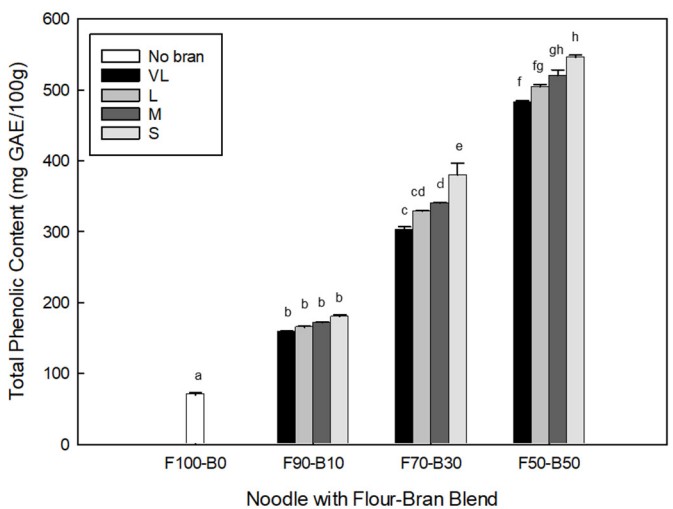
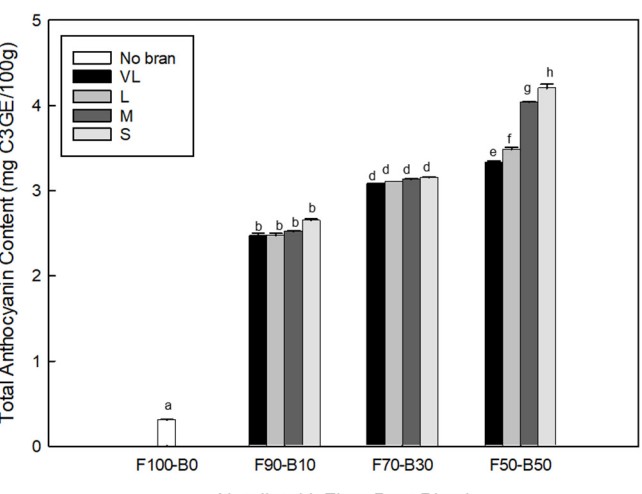

**Figure 3.** Total phenolic and anthocyanin content of fresh noodles made with flour-bran blends at different blending ratios and bran sizes. The different letters above the bars are significantly different ($p < 0.05$) according to Tukey's HSD test. VL, L, M, and S indicate the brans with very large, large, medium, and small particle sizes.

The total anthocyanin contents of the fresh noodles prepared with the blends with VL bran were 2.47 for F90-B10, 3.08 for F70-B30, and 3.33 mg C3GE/100 g for F50-B50, increasing significantly as the blending ratio increased ($p < 0.05$). Within the same blending ratio of bran, the total anthocyanin content of the noodles with blends increased slightly or significantly (e.g., from 3.10 to 3.15 mg C3GE/100 g for F70-B30 and from 3.48 to 4.20 mg C3GE/100 g for F50-B50) as bran size decreased from L to S ($p < 0.05$). Brewer et al. [56] reported that whole wheat bran showed an increase in phenolic acids and anthocyanins with decreasing particle size distribution compared to un-milled bran, showing a similar trend to the results of our study. The increase in total anthocyanin with decreasing bran size is also possibly explained by increasing surface area and extraction as the increase in TPC [56]. Like the total phenolic content, Avarzed et al. [29] reported higher total anthocyanin content in purple-colored wheat bran than in normal wheat bran.

### 3.7. Antioxidant Activity of Fresh Noodles Prepared with Flour-Bran Blends

The ABTS and DPPH radical-scavenging activities of fresh noodles prepared from flour-bran blends with different blend ratios and bran sizes are shown in Figure 4. Fresh noodles prepared with flour alone exhibited ABTS radical-scavenging activity of 78.7 μmol

TE/100 g. The ABTS radical-scavenging activity of the fresh noodles with the blends with VL bran was 245.5 for F90-B10, 315.3 for F70-B30, and 487.4 μmol TE/100 g for F50-B50, increasing significantly as the blending ratio increased ($p < 0.05$). Even with 10% blending of bran, the ABTS radical-scavenging activity of fresh noodles increased dramatically. Within the same blending ratio of bran, the ABTS radical-scavenging activity of the noodles with blends increased significantly (e.g., from 499.2 to 561.2 μmol TE/100 g for F50-B50) as the bran size decreased from L to S ($p < 0.05$). In addition, as the blending ratio increased, the increase in ABTS radical-scavenging activity of fresh noodles with reduced bran size increased. In particular, fresh noodles prepared with the blends at 50% blending ratio (F50-B50) from L to S bran showed the largest increase in ABTS radical-scavenging activity (Figure 4).

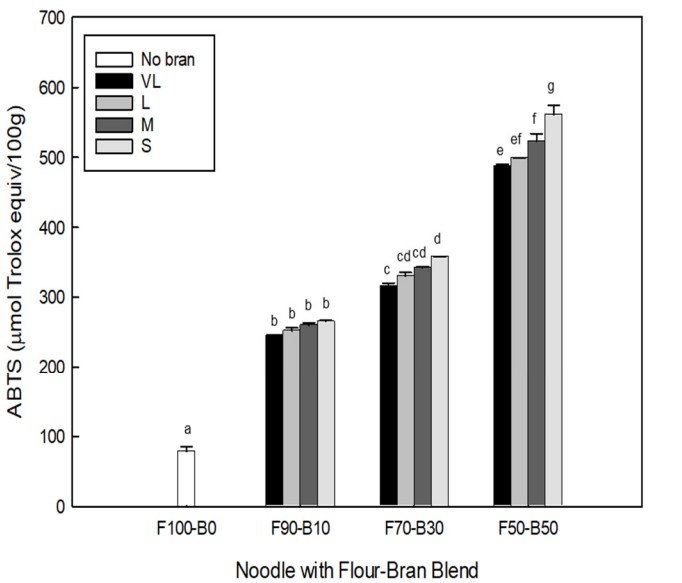
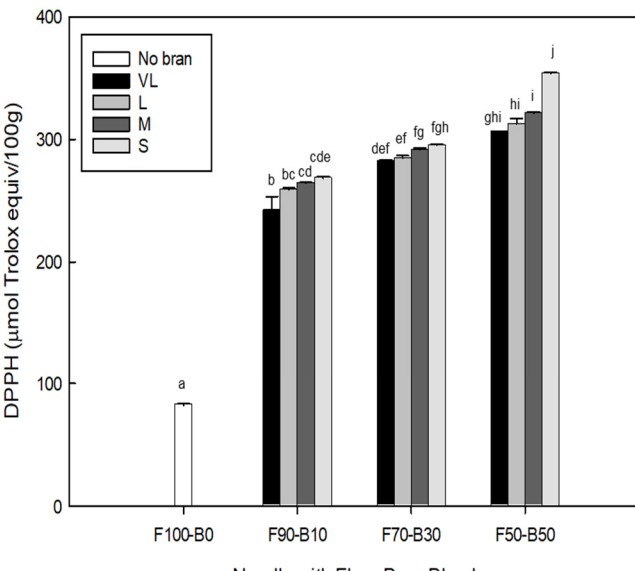

**Figure 4.** Antioxidant activity as indicated by ABTS and DPPH radical-scavenging activity of fresh noodles with flour-bran blends at different blending ratios and bran sizes. The different letters above the bars are significantly different ($p < 0.05$) according to Tukey's HSD test. VL, L, M, and S indicate the brans with very large, large, medium, and small particle sizes.

DPPH radical-scavenging activity of fresh noodles prepared with flour alone was 82.4 μmol TE/100 g. The noodles prepared with the blends containing VL bran showed the DPPH radical-scavenging activity was 242.0 for F90-B10, 282.1 for F70-B30, and 306.1 μmol TE/100 g for F50-B50, increasing significantly as the blending ratio increased ($p < 0.05$). The DPPH radical-scavenging activity of the noodles even with 10% blending of bran increased dramatically. Within the same blending ratio of bran, the DPPH radical-scavenging activity of the noodles with the blends increased slightly (e.g., from 312.5 to 353.8 μmol TE/100 g for F50-B50) as the bran size decreased from L to S ($p < 0.05$). Similar to the results of ABTS radical-scavenging activity, the extent of increase in DPPH radical-scavenging activity of fresh noodles with reducing bran size increased with increasing bran blending ratio (Figure 4). In particular, fresh noodles prepared with blends at 50% blending ratio (F50-B50) from L to S bran showed the largest increase in DPPH radical-scavenging activity. A positive correlation was found between antioxidant activity and the blending ratio of bran and bran size.

According to Avarzed et al. [29], purple-colored wheat bran has significantly higher antioxidant activity than normal wheat bran (1247 vs. 972 mg μmol TE/100 g for ABTS, 936 vs. 749 mg μmol TE/100 g for DPPH). Phenolic acids, pre-dominantly ferulic acid, are concentrated in the bran fraction of whole wheat and contribute to antioxidant properties [5,57,58]. Extracts with the highest TPC showed the greatest antioxidant properties [56]. Habuš et al. [39] also reported the highest TPC and AO in fine bran samples are possibly led by

the enhanced release of phenolic compounds from the wheat bran matrix. In our study, the enhancing effect of small bran particles on antioxidant activity appeared more significant for higher blending ratios by a relatively increased cumulative release.

## 4. Conclusions

This study demonstrated that the particle size of wheat bran has a significant impact on fresh-noodle texture and antioxidant activity. Compared to VL wheat bran, the R/E ratios of fresh noodles increased significantly with decreasing bran size. The texture of cooked noodles showed an increase in firmness and springiness upon reducing the size of the bran particles. The antioxidant activity of fresh noodles increased significantly as the bran blending ratio and bran particle size decreased. Fine bran particles were embedded more closely to the noodle sheet, resulting in fewer bran particles leaching out of the cooking water. Noodle quality and antioxidant activity were more significantly enhanced by small bran particles at higher blending ratios. Thus, reducing the bran particle size could improve the textural properties of noodles with flour-bran blends.

**Author Contributions:** Conceptualization, K.K. and M.K.; methodology, G.P. and H.C.; formal analysis, G.P. and H.C.; investigation, G.P., H.C. and M.K.; data curation, G.P. and H.C.; writing—original draft, G.P. and M.K.; validation, K.K. and M.K.; visualization, K.K. and M.K.; writing—review and editing, K.K. and M.K.; resources, M.K.; supervision, M.K.; project administration, M.K.; funding acquisition, M.K. All authors have read and agreed to the published version of the manuscript.

**Funding:** This research was supported by the Cooperative Research Program for the Agriculture Science and Technology Department (Project No. PJ014543), funded by the Rural Development Administration (Korea).

**Data Availability Statement:** The data presented in this study are available on request from the corresponding author.

**Conflicts of Interest:** The authors declare no conflict of interest.

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
