# Peer review of "Quality Characteristics and Antioxidant Activity of Fresh Noodles Formulated with Flour-Bran Blends Varied by Particle Size and Blend Ratio of Purple-Colored Wheat Bran"

_processes, doi:10.3390/pr10030584_

Round 1

Reviewer 1 Report

  1. Table 1: The amount of water added for different recipes is different. Please explain how to determine the amount of water to be added for different formulations? In addition, the total moisture content of the different formulations should be listed.
  2. In general, cooking loss is a more direct and accurate method for assessing cooking loss of noodle than turbidity of cooking water. Why did this study choose turbidity as the index for the cooking loss of noodles?
  3. Line 111: The unit of particle size is wrong.
  4. Line 114, 249, 251 & 258: “VF bran” should be VL bran.
  5. Figure 1: The quality of photo should improve.
  6. Table 6 & line 376-378: In this paragraph, why compare the difference in the texture of the noodles of the “fresh” control sample (F100-B0) and the “cooked” noodles of the experimental samples (flour-bran blends)? What does it mean?
  7. Line 336-337: How to get this inference? It should be explained in detail.

Author Response

We appreciate the detailed comments. We revised the manuscript marked with red color.

  1. Table 1: The amount of water added for different recipes is different. Please explain how to determine the amount of water to be added for different formulations? In addition, the total moisture content of the different formulations should be listed.

Ans) In Line 142-143, it is explained that the amount of water (30–43.3 g in Table 1) was added based on the water absorption capacity measured by the water SRC value of each blend.

  1. In general, cooking loss is a more direct and accurate method for assessing cooking loss of noodle than turbidity of cooking water. Why did this study choose turbidity as the index for the cooking loss of noodles?

Ans) Although analyzing the cooking loss of noodles is commonly used, turbidity of cooking water is also measured to assess cooking loss of noodles easily and fast.

  1. Line 111: The unit of particle size is wrong.

Ans) The unit was corrected.

  1. Line 114, 249, 251 & 258: “VF bran” should be VL bran.

Ans) Corrected from VF bran to VL bran.

  1. Figure 1: The quality of photo should improve.

Ans) The photo has been replaced.

  1. Table 6 & line 376-378: In this paragraph, why compare the difference in the texture of the noodles of the “fresh” control sample (F100-B0) and the “cooked” noodles of the experimental samples (flour-bran blends)? What does it mean?

Ans) The texture of the noodles was measured on cooked noodles. However, it was found that the expression “fresh” was unclear, and the sentence was corrected.

  1. Line 336-337: How to get this inference? It should be explained in detail.

Ans) We revised the sentence, including an explanation.

Reviewer 2 Report

The authors incorporated varied ratios of purple-colored wheat bran with different particle sizes into noodle making. They found that flour blended with smaller-sized brans at higher blending ratios showed better noodle-making performance, when compared with larger-sized brans. These results are already known. I suggest a major revision to improve the quality of this manuscript.

1. Give detailed method for noodles' appearance measurement.
2. Because the reported effect of the bran particle size on the dough rheological property is contradictory in literature, it is not suitable to cite the conclusions from other's study. I recommend the authors to find out rheology properties of their own dough.
3. Please explain why F50-B50 noodles quality seemed better than the  whole flour noodles for lower cooking loss and higher hardness
?Especially, the S bran contains the highest amount of damaged starch. As the authors interpreted the damaged starches may increase the cooking loss and make the cooked noodle become gummier and less firm.
4. It better to measure the antioxidant activity of cooked noodles.
5. A high antioxidant activity resulted in low storage stability. I recommend the authors to estimate the shelf life of the noodles.
6. English should be improved.

Author Response

The authors incorporated varied ratios of purple-colored wheat bran with different particle sizes into noodle making. They found that flour blended with smaller-sized brans at higher blending ratios showed better noodle-making performance, when compared with larger-sized brans. These results are already known. I suggest a major revision to improve the quality of this manuscript.

We appreciate the detailed comments. We revised the manuscript marked with blue color.

  1. Give detailed method for noodles' appearance measurement.

Ans) We added the details for the method. 

  1. Because the reported effect of the bran particle size on the dough rheological property is contradictory in literature, it is not suitable to cite the conclusions from other's study. I recommend the authors to find out rheology properties of their own dough.

Ans) We revised the sentences.

  1. Please explain why F50-B50 noodles quality seemed better than the whole flour noodles for lower cooking loss and higher hardness? Especially, the S bran contains the highest amount of damaged starch. As the authors interpreted the damaged starches may increase the cooking loss and make the cooked noodle become gummier and less firm.

Ans) We did not test the noodles prepared with whole wheat flour. Control noodle was the noodles prepared with refined ‘Keumkang’ flour. Although the damaged starch content increased by decreasing bran size, the developed gluten network appeared a more significant impact on cooking loss and texture of cooked noodles.  

  1. It better to measure the antioxidant activity of cooked noodles.

Ans) We agree partly with the comment, but we considered measuring the antioxidant activity of fresh noodles is also essential to estimate the change in the activity during baking, which will perform another study in the future.  

  1. A high antioxidant activity resulted in low storage stability. I recommend the authors to estimate the shelf life of the noodles.

Ans) I expect the high antioxidant activity of food generally is resulted in high storage stability. In this study, we prepared fresh noodles, which usually and often cooked and consumed immediately in Korea, to focus on potential application of purple-colored wheat bran. Of course, the shelf life of the fresh commercial noodles in the market depends on the processing method and packaging. Therefore, it is not easy to estimate the shelf life of the noodles prepared in this study.  

  1. English should be improved.

Ans) The manuscript has been already reviewed and revised by professional editing service.

Reviewer 3 Report

The manuscript has explored the effect of different-particle-size and purple-colored wheat bran on characteristics and antioxidant activity of noodle. This manuscript is interesting. The comments were listed below.

Table and Figure legend: Table and Figure legends should be written appropriately so that readers can understand without reading the text. When specific or unusual abbreviations are used, they shall appear in brackets after each complete term the first time they are used, such as F and B in table 1; L, M, S in Figure 1; VL, L, M, S in Table 2; SRC etc.  

There are two Fig. 1. The author used VF in the text (L 249, 251) but VL in tables 1-6.

L342-344; bread dough?

L443-451; The authors do not explain appropriately why anthocyanin content increased.

L490-491; What is the reason for the increased antioxidant activity? It is unlikely that the smaller particle size is the reason.

L502-505; I think this sentence does not deserve to be the conclusion.

Author Response

The manuscript has explored the effect of different-particle-size and purple-colored wheat bran on characteristics and antioxidant activity of noodle. This manuscript is interesting. The comments were listed below.

We appreciate the detailed comments. We revised the manuscript marked with green color.

Table and Figure legend: Table and Figure legends should be written appropriately so that readers can understand without reading the text. When specific or unusual abbreviations are used, they shall appear in brackets after each complete term the first time they are used, such as F and B in table 1; L, M, S in Figure 1; VL, L, M, S in Table 2; SRC etc.  

Ans) Added the notes in Tables and Figures.

There are two Fig. 1. The author used VF in the text (L 249, 251) but VL in tables 1-6.

Ans) Corrected the number of Fig 2 from Fig 1.

L342-344; bread dough?

Ans) Corrected to “noodle dough.”

L443-451; The authors do not explain appropriately why anthocyanin content increased.

Ans) Explanation was added.

L490-491; What is the reason for the increased antioxidant activity? It is unlikely that the smaller particle size is the reason.

Ans) Explanation was added.

L502-505; I think this sentence does not deserve to be the conclusion.

Ans) The sentence was omitted.

Round 2

Reviewer 1 Report

No.

Reviewer 2 Report

Revisions are acceptable.